

# Challenging a bioinformatic tool's ability to detect microbial contaminants using *in silico* whole genome sequencing data

Nathan D. Olson, Justin M. Zook, Jayne B. Morrow and Nancy J. Lin

Material Measurement Laboratory, National Institute of Standards and Technology, Gaithersburg, MD, United States of America

## ABSTRACT

High sensitivity methods such as next generation sequencing and polymerase chain reaction (PCR) are adversely impacted by organismal and DNA contaminants. Current methods for detecting contaminants in microbial materials (genomic DNA and cultures) are not sensitive enough and require either a known or culturable contaminant. Whole genome sequencing (WGS) is a promising approach for detecting contaminants due to its sensitivity and lack of need for *a priori* assumptions about the contaminant. Prior to applying WGS, we must first understand its limitations for detecting contaminants and potential for false positives. Herein we demonstrate and characterize a WGS-based approach to detect organismal contaminants using an existing metagenomic taxonomic classification algorithm. Simulated WGS datasets from ten genera as individuals and binary mixtures of eight organisms at varying ratios were analyzed to evaluate the role of contaminant concentration and taxonomy on detection. For the individual genomes the false positive contaminants reported depended on the genus, with *Staphylococcus*, *Escherichia*, and *Shigella* having the highest proportion of false positives. For nearly all binary mixtures the contaminant was detected in the *in-silico* datasets at the equivalent of 1 in 1,000 cells, though *F. tularensis* was not detected in any of the simulated contaminant mixtures and *Y. pestis* was only detected at the equivalent of one in 10 cells. Once a WGS method for detecting contaminants is characterized, it can be applied to evaluate microbial material purity, in efforts to ensure that contaminants are characterized in microbial materials used to validate pathogen detection assays, generate genome assemblies for database submission, and benchmark sequencing methods.

Corresponding author
Nathan D. Olson, nolson@nist.gov

## INTRODUCTION

Microbial materials such as cells and extracted genomic DNA from a presumably pure culture should ideally be free of organismal contaminants, yet rarely are. High sensitivity detection methods including polymerase chain reaction (PCR) and next generation sequencing (NGS) can detect organismal contaminants previously undetectable by traditional microbiological methods. Characterizing these contaminants in order to focus efforts on reducing their level is critical to ensuring high-quality microbial materials are

used to populate sequence databases (*Parks et al., 2015*), for mock communities to validate metagenomic methods (*Bokulich et al., 2016*), to validate biodetection assays (*Ieven, Finch & van Belkum, 2013*; *Coates, Brunelle & Davenport, 2011*), and for basic research using model systems (*Shrestha et al., 2013*). Furthermore, tools to assess general contaminants are also needed for the characterization of microbial genomic reference materials (*Olson et al., 2016*), where contaminant profiles allow users to properly determine whether the material is suitable for their application. Contaminants in microbial materials, as found in non-axenic cellular materials or genomic materials with foreign DNA, have been addressed when processing the sequencing data but not for general material characterization (*Shrestha et al., 2013*; *Tennessen et al., 2015*). PCR and NGS can also detect reagent impurities. Reagent contaminants can be addressed by producing negative controls (*Jervis-Bardy et al., 2015*), improved methods for removing contaminants (*Woyke et al., 2011*; *Motley et al., 2014*), and post-processing of sequence data in effort to distinguish these impurities from true organismal contaminant in the microbial material (*Mukherjee et al., 2015*).

Current approaches for detecting contaminants in microbial materials such as culture, microscopy, or PCR typically fail to meet all the requirements to characterize microbial materials for routine applications. Culture- and microscopy-based methods lack the required sensitivity for detecting contaminants in microbial materials being used in NGS and PCR applications, are not appropriate for genomic DNA materials, and assume the contaminants are phenotypically distinct from the material they contaminate. While PCR-based methods can detect contaminants in genomic DNA, the methods are limited to specifically targeted contaminants and are not amenable to highly multiplexed applications (*Heck et al., 2016*; *Marron, Akam & Walker, 2013*). In contrast to these methods, shotgun metagenomic methods, though unable to assess contaminant viability, can be used to detect contaminants in both cell cultures and genomic DNA materials while only requiring the contaminant has sequencing reads differentiating it from the material strain. As whole genome sequencing can be performed on genomic DNA and cell cultures (after DNA extraction), the method is appropriate for both types of microbial materials.

Shotgun metagenomic sequencing is used to characterize environmental samples, detect pathogens in clinical samples, and is suitable for detecting contaminants in microbial materials. Shotgun metagenomics consists of two main steps, whole genome sequencing of all DNA in a sample, and analysis of the resulting sequencing data, most commonly using a taxonomic assignment algorithm (*Thomas, Gilbert & Meyer, 2012*). For genomic DNA materials, the material itself is sequenced, whereas for cells the genomic DNA must first be extracted from cell cultures prior to sequencing. After sequencing, a taxonomic assignment algorithm is used to characterize the sequencing data. Currently, researchers use a variety of classification algorithms with varying accuracy and computational performance (*Bazinet & Cummings, 2012*; *Menzel, Ng & Krogh, 2016*; *Sczyrba et al., 2017*). Nearly all methods require a reference database, where the contaminating organism (or an organism more closely related to the contaminant than the material) must be in the database for it to be detected. Bioinformatic methods have been developed and used to detect contaminant reads in a whole genome sequencing dataset but to our knowledge have not been used to detect contaminants in a microbial material (*Kumar et al., 2013*; *Delmont & Eren, 2016*).
In order to confidently use metagenomics to detect contaminants in microbial materials, one must first understand its limitations in doing so. We have developed a metagenomics-based approach to evaluate contaminant detection capabilities. In this work, we present results from an *in-silico* study demonstrating our approach using an existing taxonomic assignment algorithm for detecting contaminant DNA in simulated microbial whole genome sequence data. First, a baseline assessment of the method was performed using simulated sequencing data from single microorganisms to characterize the types of false positive contaminants the algorithm may report. The contaminant detection method was then evaluated for its ability to detect organismal contaminants in microbial material strains using sequencing data simulated to replicate microbial materials contaminated with different organismal contaminants at a range of concentrations.

This manuscript is intended for users and maintainers of microbial material stocks who are interested in validating material purity and understanding the limitations of their validation method. A secondary audience is taxonomic classification algorithm developers, as this work presents a novel approach to evaluating taxonomic classification methods and an additional use case that developers may not have previously considered.

## METHODS

Simulated whole genome sequence data and metagenomic taxonomic classification methods were used to detect and identify foreign DNA in microbial materials (genomic DNA and cultures). Simulated data from individual prokaryotic genomes were used to characterize how well the method correctly classifies reads at the species level. To evaluate contaminant detection we used datasets comprised of pairwise combinations of simulated reads from individual genomes.

### Simulation of sequencing data

To approximate real sequencing data, reads were simulated using an empirical error model and insert size distribution. Whole genome sequence data were simulated using the ART sequencing read simulator (*Huang et al., 2012*). Reads were simulated with the Illumina MiSeq error model for $2 \times 230$ base pair (bp) paired-end reads with an insert size of $690 \pm 10$ bp (average $\pm$ standard deviation) and 20 X mean coverage. The insert size parameters were defined based on the observed average and standard deviation insert size of the NIST RM8375-MG002 MiSeq sequencing data (*Olson et al., 2016*) (NCBI Biosample accession SAMN02854573).

### Assessment of taxonomic composition

The taxonomic composition of simulated datasets was determined using the PathoScope sequence taxonomic classifier (*Francis et al., 2013*). PathoScope was selected for two reasons: (1) it uses a large reference database reducing potential biases due to contaminants not represented in the database, and (2) it leverages efficient whole genome read mapping algorithms. Additionally, PathoScope was successfully used in our pilot study (https://doi.org/10.6084/m9.figshare.1200090.v1) and as part of the pipeline developed to characterize the NIST microbial genomic DNA reference

**Table 1 Breakdown of the number of genomes by genus used to generate single genome simulated datasets.** *N* indicates the number of genomes (406 total), and Genome Size is presented as the median and range (minimum to maximum). Species indicates the number of different species for each genus included in the baseline assessment.

| Genus | N | Species | Genome size (Mb) |
|---|---|---|---|
| *Bacillus* | 76 | 19 | 5.05 (3.07–7.59) |
| *Clostridium* | 32 | 15 | 4.02 (2.55–6.67) |
| *Escherichia* | 62 | 1 | 5.11 (3.98–5.86) |
| *Francisella* | 18 | 4 | 1.89 (1.85–2.05) |
| *Listeria* | 39 | 5 | 2.97 (2.78–3.11) |
| *Pseudomonas* | 57 | 21 | 6.18 (4.17–7.01) |
| *Salmonella* | 44 | 2 | 4.88 (4.46–5.27) |
| *Shigella* | 10 | 4 | 4.74 (4.48–5.22) |
| *Staphylococcus* | 49 | 2 | 2.82 (2.69–3.08) |
| *Yersinia* | 19 | 3 | 4.73 (4.62–4.94) |

material (Olson et al., 2016). This method uses an expectation maximization algorithm where the sequence data are first mapped to a database comprised of all sequence data in the Genbank nt database. Then, through an iterative process, it re-assigns ambiguously mapped reads based on the proportion of reads mapped unambiguously to individual taxa in the database. The PathoScope 2.0 taxonomic read classification pipeline has three steps; (1) PathoQC—read quality filtering and trimming using the PRINSEQ algorithm (Schmieder & Edwards, 2011), (2) PathoMap—mapping reads to a reference database using the bowtie2 algorithm (Langmead & Salzberg, 2012), and (3) PathoID—expectation–maximization classification algorithm. The annotated Genbank nt database provided by the PathoScope developers was used as the reference database (ftp://pathoscope.bumc.bu.edu/data/nt_ti.fa.gz).

## Baseline assessment using individual genomes

Simulated sequencing data from individual genomes was used to characterize the false positive contaminants reported by PathoScope. Sequence data was simulated for 406 strains, from ten genera (Table 1, Table S1). These genera were selected based on relevance to public health and biothreat detection. We will refer to the genome used to generate the reads as the material genome. The genomes included in the simulation study were limited to closed genomes in the Genbank database (http://www.ncbi.nlm.nih.gov/genbank/, accessed 10/18/2013) belonging to genera of interest (Table 1). Due to the large number of *Escherichia* and *Staphylococcus* genomes, genomes from these genera were limited to the species *Escherichia coli*, and *Staphylococcus aureus* respectively. We note that after the genomes were selected one of the *Staphylocuccus aureus* genomes was renamed *S. argenteus*, Genbank taxid 985002 (Tong et al., 2015). Average nucleotide identity for all pair-wise comparisons was calculated using MUMMER3 and the `ani_pairs.R` script in the project github repository (Kurtz et al., 2004). The taxonomic hierarchy for the material genome and simulated read assignment match levels were determined using the R package, Taxize (Chamberlain & Szöcs, 2013; Chamberlain et al., 2016).

**Table 2  Representative strains used in simulated contaminant datasets, based on available type strains.** Match proportion indicates the estimated proportion of the material assigned to the correct species by PathoScope. Aligned Reads is the number of simulated reads aligned to the database by PathoMap. DNA size is the total size of the genome, chromosome and plasmids in Mb.

| Representative strain | Match proportion | Aligned reads | Mb |
|---|---|---|---|
| *Bacillus anthracis* str. Ames | 1.00 | 227,270 | 5.23 |
| *Clostridium botulinum* A str. Hall | 1.00 | 163,500 | 3.76 |
| *Escherichia coli* O157:H7 str. EC4115 | 0.98 | 247,990 | 5.70 |
| *Francisella tularensis* subsp. *tularensis* SCHU S4 | 1.00 | 82,290 | 1.89 |
| *Pseudomonas aeruginosa* PAO1 | 1.00 | 272,360 | 6.26 |
| *Salmonella enterica* subsp. *enterica* serovar Typhimurium str. D23580 | 1.00 | 212,140 | 4.88 |
| *Staphylococcus aureus* subsp. *aureus* ED133 | 0.98 | 123,150 | 2.83 |
| *Yersinia pestis* CO92 | 1.00 | 209,970 | 4.83 |

## Contaminant detection assessment

Simulated contaminated datasets were used to evaluate how contaminant detection varied by material and contaminant genome over a range of contaminant concentrations. Representative genomes for eight of the 10 genera were used to generate the simulated contaminant datasets (Table 2, Table S2). An *Escherichia coli* strain was selected as a representative of both *Escherichia* and *Shigella*, as the genus *Shigella* and species *Escherichia coli* are not phylogenetically resolved (*Lan & Reeves, 2002*). No representative genome for *Listeria* was included in this part of the study. For each pairwise combination of representative genomes, the simulated contaminant dataset was comprised of a randomly selected subset of reads from the material and contaminant (Fig. 1). The simulated datasets were randomly subsampled at defined proportions, with $p$ representing the proportion of reads from the contaminant, and $1 - p$ the proportion of reads from the material dataset. A range of contaminant proportions at 10-fold increments was simulated with $p$ ranging from $10^{-1}$ to $10^{-8}$, resulting in 512 simulated contaminant datasets. This approach simulates the proportion of cells in a contaminated material and not the amount of DNA, assuming unbiased DNA extraction. Organisms with larger genomes, therefore, have more simulated reads.

To generate the simulated contaminant datasets, single organism simulated datasets were first generated for the 8 representative genomes using the same methods as baseline assessment (Fig. 1 and Table 2). The resulting simulated sequencing data was first processed using the PathoQC and PathoMap steps in the PathoScope pipeline. The output from the PathoMap step (SAM file, sequence alignment file: https://samtools.github.io/hts-specs/SAMv1.pdf) for the material and contaminant datasets were subsampled as described above then combined. The resulting SAM file was processed by PathoID, the third step in the PathoScope pipeline. Subsampling the SAM files instead of the simulated sequence files greatly reduces the computational cost of the analysis, as the simulated reads were only processed once by the first two steps in the PathoScope pipeline rather than for every simulated contaminant dataset. For simulated datasets with contaminant proportions

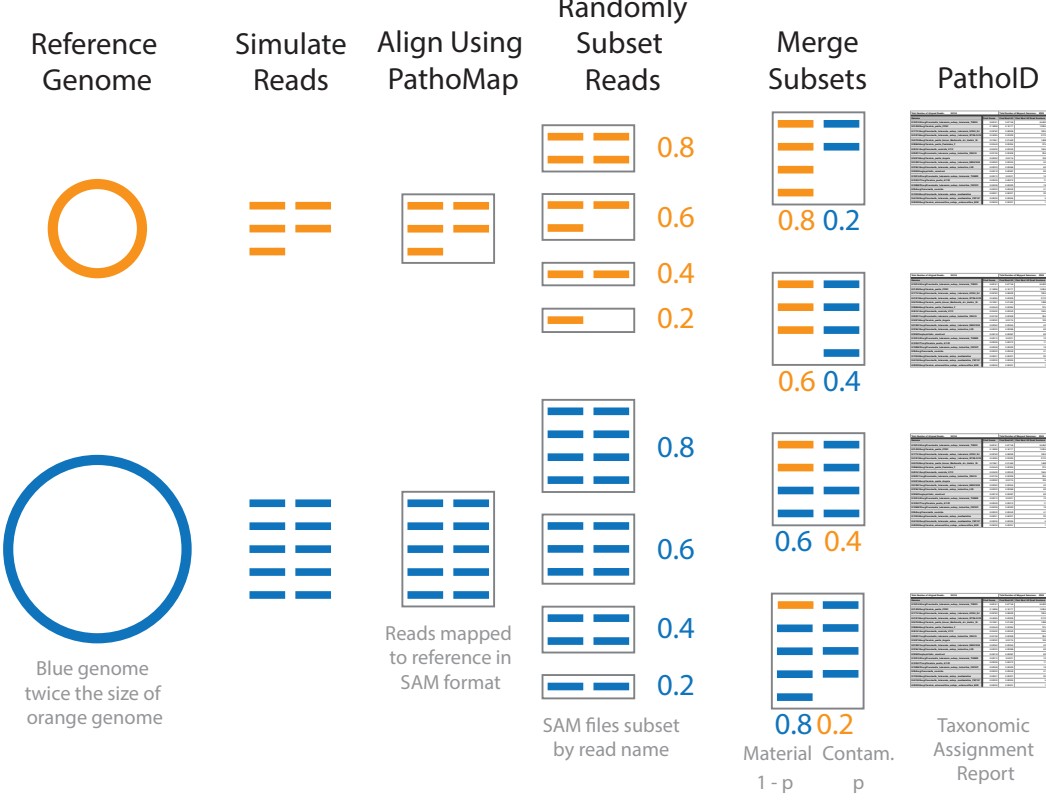

**Figure 1** **Diagram of simulated contaminant dataset workflow for two individual genomes.** Contaminant proportions (p) of 0.2 and 0.4 are used for demonstration purposes. The reads were initially simulated from individual genomes. The blue genome is twice the size of the orange genome, and twice as many reads are simulated for the blue genome compared to the orange in order to obtain the same coverage. The simulated reads were aligned to the reference database using PathoMap. The resulting alignment file, in SAM file format, was randomly subset based on the desired proportions. Complementary subsets of SAM files (e.g., 0.8 material and 0.2 contaminant) from the two genomes were merged to create individual simulated contaminant datasets. Due to the different sized genomes, the simulated contaminant datasets have different numbers of reads. Taxonomic assignment summary tables were generated from simulated contaminant datasets using PathoID.

greater than $10^{-5}$, the quantitative accuracy of the contaminant detection method was assessed by comparing the defined contaminant proportion (true proportion) to the PathoScope contaminant proportion (estimated proportion). Pearson's correlation coefficient was used to evaluate agreement between the true and estimated proportions. The error rate, *(estimated−true)/true*, was compared across material and contaminant combinations.

## Bioinformatics pipeline

To facilitate repeatability and transparency, a Docker (http://www.docker.com) container is available with pre-installed pipeline dependencies (https://hub.docker.com/r/natedolson/docker-pathoscope/). The scripts used to run the simulations are available at https://github.com/nate-d-olson/genomic_purity. Additionally, seed numbers for the random number generator were randomly assigned and recorded

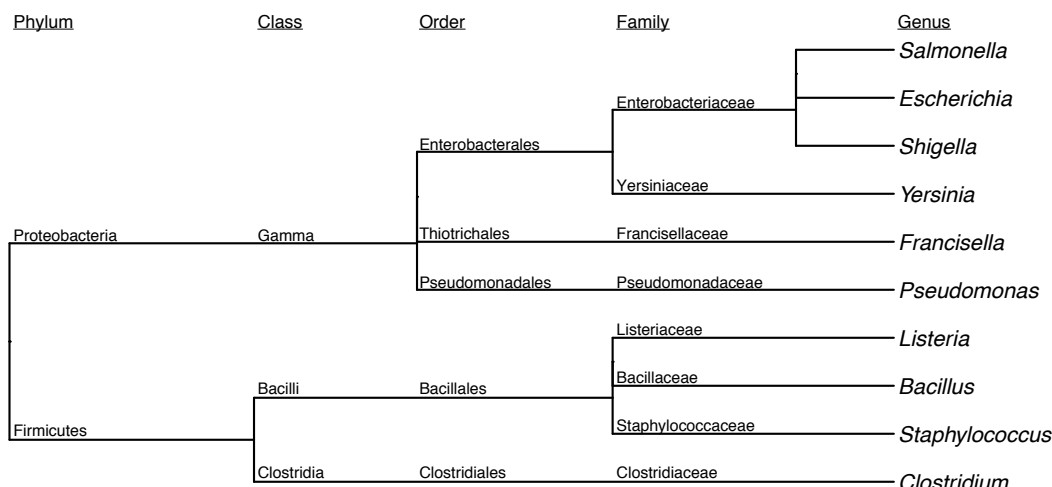

**Figure 2** Dendrogram depicting the taxonomic lineage of genera used in baseline assessment.

for each dataset so the simulated datasets used in the study could be regenerated. PathoScope results were processed and analyzed using the statistical programming language R (*R Core Team, 2016*), and intermediate analysis and data summaries were organized using ProjectTemplate (*White, 2014*) and archived in a GitHub repository (https://github.com/nate-d-olson/genomic_purity_analysis) along with the source files for this manuscript.

## RESULTS

### Baseline assessment using individual genomes

First, we assessed the baseline performance of the proposed contaminant detection method. We applied our method to simulated sequencing data from individual genomes. All reads assigned to a different taxa than the genome the reads were simulated from were defined as false positive contaminants. (This assumes the genome sequence is contaminant free.) Our analysis included taxonomic classification results for simulated sequencing data from 406 genomes, representing 10 different genera (Table 1, Fig. 2, Table S1). The 10 genera were from the Gammaproteobacteria Class and Firmicutes phylum with 1 to 21 species from each genera representing a range of genomic similarity within the genus (Fig. 3). For the genomes included in the study, *Escherichia*, *Shigella*, *Salmonella*, and *Staphylococcus* have higher within genus similarity whereas *Clostridium* and *Pseudomonas* had lower within genus similarity.

The taxonomic classification method was evaluated using the estimated proportion of species level matches. The estimated match proportion is the sum of the Final Guess values, proportions reported by PathoScope for a taxa, for all correct species level matches. For 301 of the 406 genomes, PathoScope estimated that greater than 0.99 of the material was the expected species (Fig. 4). Of the remaining 105 genomes, the estimated proportion identified as the correct species varied by material genus. All of the *Shigella* genomes and only 44 of the 49 *Staphylococcus* genomes had estimated proportions for the correct
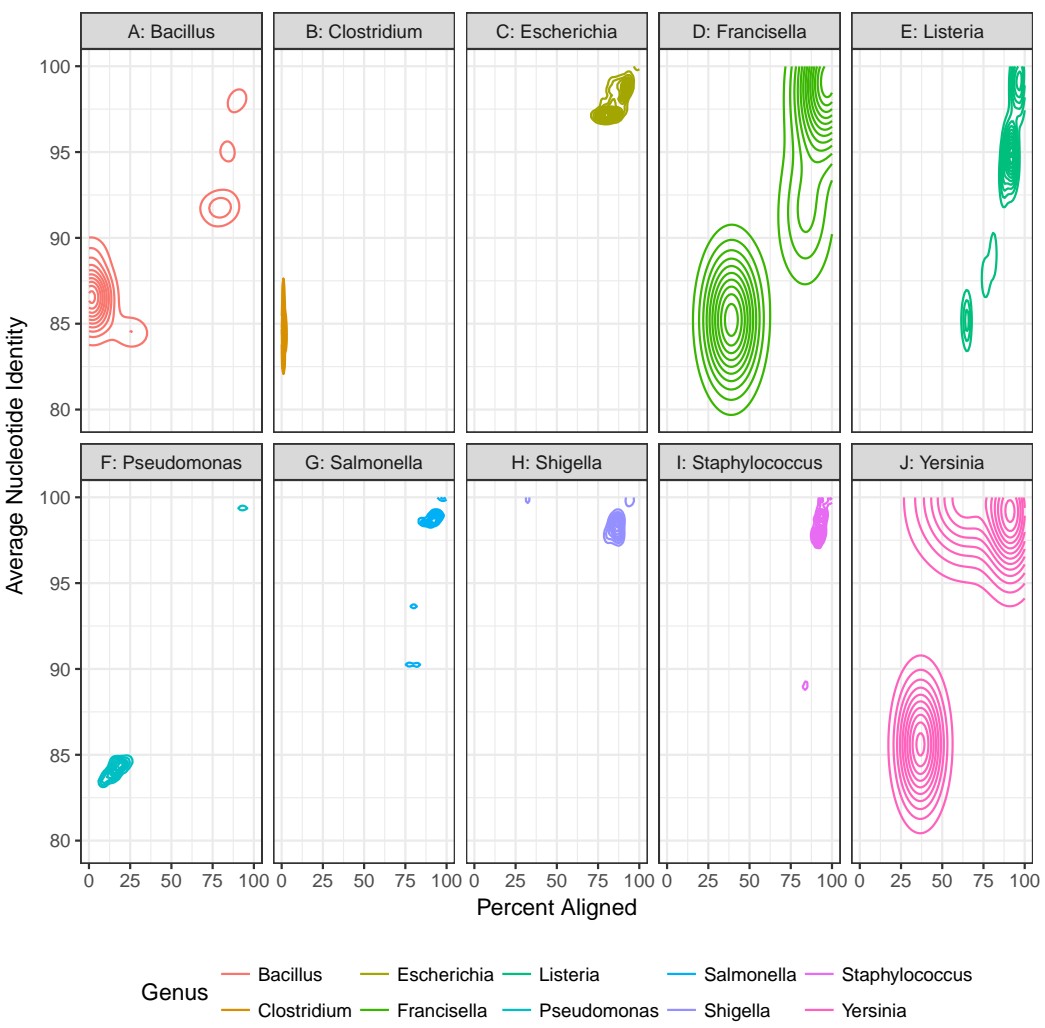

**Figure 3** **Genomic diversity of strains used in baseline study by genus.** The percent of the genome aligned is represented on the *x*-axis with average nucleotide identity (similarity of aligned regions) on the *y*-axis. More similar genomes will have a higher percent aligned and average nucleotide identity.

species less than than 0.9. 87 of those 105 genomes come from *Shigella*, *Staphylococcus*, or *Escherichia*. Excluding *Shigella*, *Escherichia*, and *Staphylococcus*, the median estimated proportion matching at the species level or higher is 0.9996. We characterized false positive contaminants detected in genomes from the genera *Shigella*, *Escherichia*, and *Staphylococcus*, as well as genomes of other species with match proportions less than 0.9. Two types of false positive contaminants were identified: (1) contaminants that were genomically indistinguishable from the material and (2) contaminants due to errors in the reference database. Sequences can be genomically indistinguishable due to phylogenetic relatedness of the organisms as well as the transfer of sequences horizontally transferred between organisms such as plasmids and genes involved in horizontal gene transfer events (*Shintani, Sanchez & Kimbara, 2015*; *Polz, Alm & Hanage, 2013*).

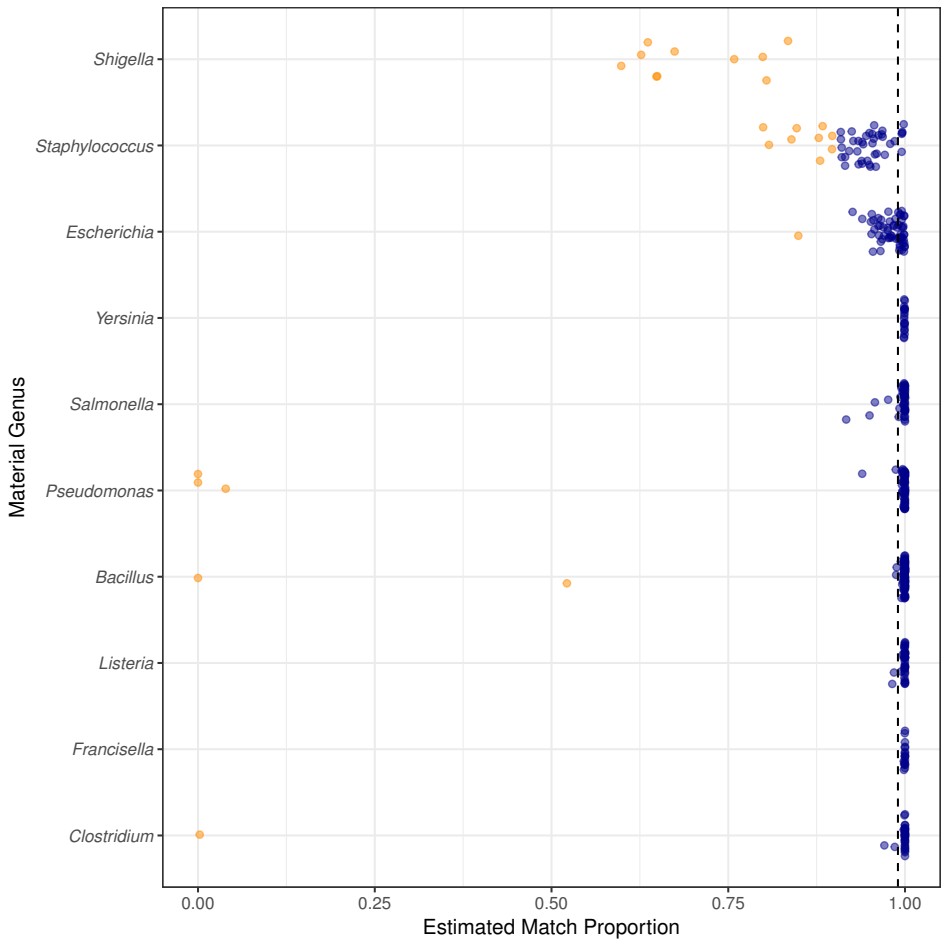

**Figure 4 Species level or higher estimated match proportion varies by material genus.** The estimated match proportion is the total proportion of the material with correct taxonomic assignments to the genome species, subspecies, strain, or isolate level. The Estimated Match Proportions shown are the Final Guess values in the PathoScope results table. Each point is calculated for a genome from a different isolate within the genus. The vertical dashed line indicates the 0.99 estimated match proportion. Orange points are genomes with species level estimated match proportions less than 0.90 and blue points greater than or equal to 0.90.

Two genomes can be genomically indistinguishable if the majority of the two genome sequences are highly similar. Phylogenetically closely related organisms are expected to have large genomic regions with high levels of similarity. Phylogenetic similarity is at least partially responsible for the low species level estimated match proportion for *Shigella* and *Escherichia*, as *Shigella* is not phylogenetically distinct from *E. coli* (*Lan & Reeves, 2002*). When including matches to *E. coli* as species level matches, the median estimated match proportions for *Shigella* genomes increases from 0.66 to 0.92. Another example of false positives at the species level due to phylogenetic similarity was low match percentage for *Clostridium autoethanogenum* strain DSM10061, where *Clostridium ljungdahlii* strain DSM13528 was assigned the top proportion of reads (0.998) instead of *C. autoenthanogenum*. False positive contaminants due to phylogenetic similarity are
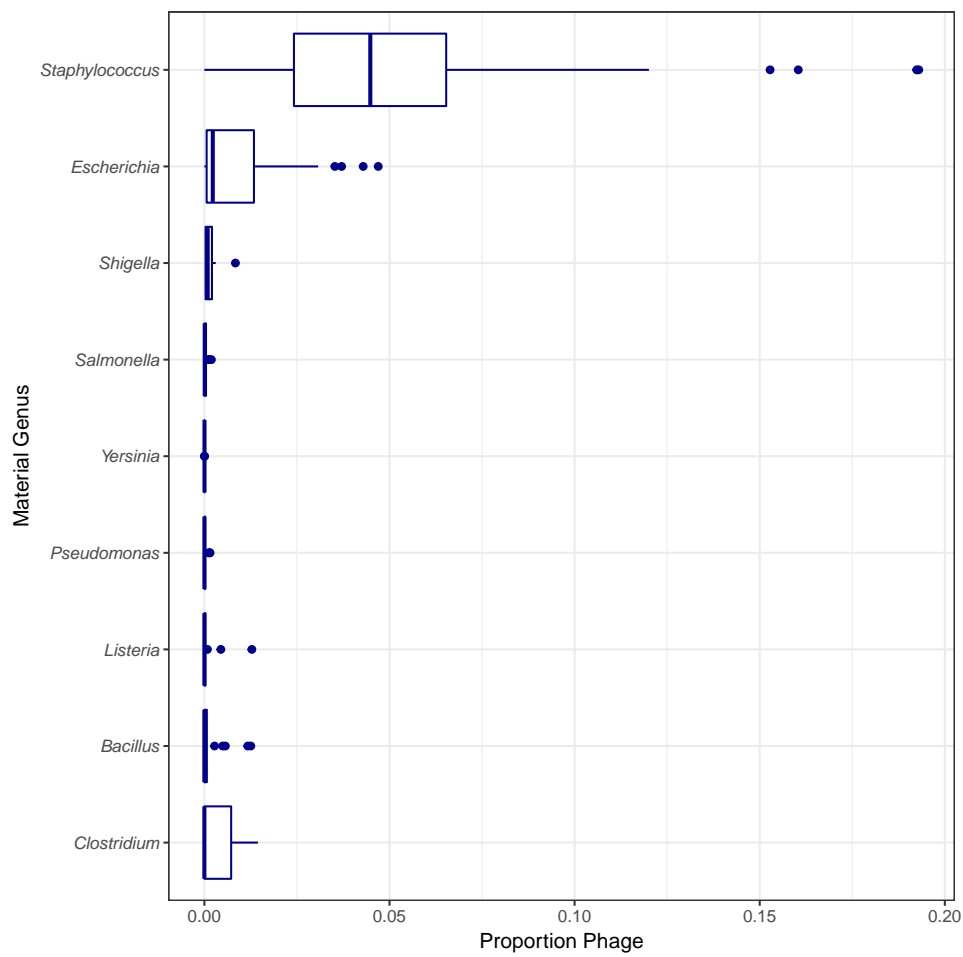

**Figure 5** **Estimated proportion of phage in the simulated single genome datasets by genera.** Final Guess values reported by PathoScope used to calculate estimated proportions. No phage were reported by PathoScope for any *Francisella* genomes.

not limited to closely related species or genus. *Escherichia coli* strain UMNK88 low match proportions were due to two bacteria in the same family as *E. coli* (Enterobacteriaceae): *Providencia stuartii* and *Salmonella enterica* subsp. *enterica* serovar Heidelberg, which had estimated match proportions of 0.11 and 0.03, respectively. False positives were also due to shared genetic material between bacteria and their phage. Phage were identified as false positive contaminants at varying proportions for genomes from all genera investigated, excluding *Francisella* (Fig. 5). The low proportions of species level matches for *E. coli* and *Staphylococcus* are partly due to relatively higher proportions of matches to phage, compared to the other genera investigated. Based on phage names, all of the false positive phage contaminants were specific to the taxonomy of the material genome.

False positive contaminants were also due to potential errors in the database such as unclassified or misclassified sequences and genome assemblies in the database containing sequence data from organismal or reagent contaminants. Low estimated match proportions can also be due to the database containing unclassified sequence data for organisms

with genomic regions that are highly similar to regions of the material genome. For example, the low estimated match proportion for *Pseudomonas* strain FGI182 was due to matches to unclassified bacteria, bacterium 142412, and unclassified *Pseudomonas* species, *Pseudomonas* sp. HF-1. The low estimated match proportion of species level matches for *Pseudomonas* strain TKP was also due to potentially misclassified sequences (*Thioalkalivibrio sulfidophilus* strain HL-EbGr7, estimated match proportion 0.0648). *Bacillus subtilis* BEST7613 genome had low species level estimated match proportion due to *Synechocystis* sp. PCC 6803 substr. PCC-P being estimated as comprising 47% of the material. *Synechocystis* is in a different phylum compared to *Bacillus* (cyanobacteria versus firmicutes) and is a false positive due to a misclassification. The *Bacillus subtilis* BEST7613 genome in the database is a synthetic chimeric genome constructed from *Bacillus subtilis* BEST7613 and *Synechocystis* sp. PCC 6803 substr. PCC-P not *Bacillus subtilis* BEST7613 (*Watanabe et al., 2012*). The *Bacillus subtilis* BEST7613 genome assembly (GenBank Accession GCA_000328745.1) was flagged by the databases curators as an anomalous assembly and removed from the RefSeq database. The genome sequences used to populate the reference database can contain contaminants themselves (*Parks et al., 2015*). These database contaminants are responsible for additional false positive contaminants. The species level estimated match proportion for *Pseudomonas* strain TKP was partially due to contaminated genome sequences in the database (wheat—*Triticum aestivum* estimated match proportion 0.087). The eukaryotic false positive contaminants are likely due to contaminants in the eukaryotic DNA extract or reagents used to generate the sequencing data for the assembly (*Parks et al., 2015*).

## Contaminant detection assessment

Finally, contaminant detection was assessed by combining subsets of simulated data from two organisms at defined proportions, with the larger proportion representing the microbial material and smaller proportion the contaminant (Fig. 1). We simulated contaminant datasets as pairwise combinations of representative genomes from eight of the genera used in the baseline assessment section of the study (Table 2, Fig. 6). All of the genomes selected have a species level estimated match proportion greater than 0.98 (Table 2). The representative genomes had low pairwise similarity based on the average nucleotide identity analysis (Fig. 6) with average identity between 82% and 86% with greater than 1% of the genomes aligned for four of the 28 organism pairs. The *Salmonella* and *E. coli* had the highest percent of their genomes aligned to each other at 36% and 30%, respectively.

The minimum contaminant proportion detected was $10^{-3}$ and $10^{-4}$ for most pairwise comparisons with a few exceptions (Fig. 7). The similarity between the material and contaminant genomes did not appear to impact the minimum contaminant proportion detected (Fig. 6). However, this is likely due to the overall low level of similarity between the representative genomes. When *Y. pestis* was the simulated contaminant, the minimum detected proportion was 0.1 for all material strains. For all simulated datasets where *F. tularensis* was the contaminant, the contaminant was not detected. It is unclear why
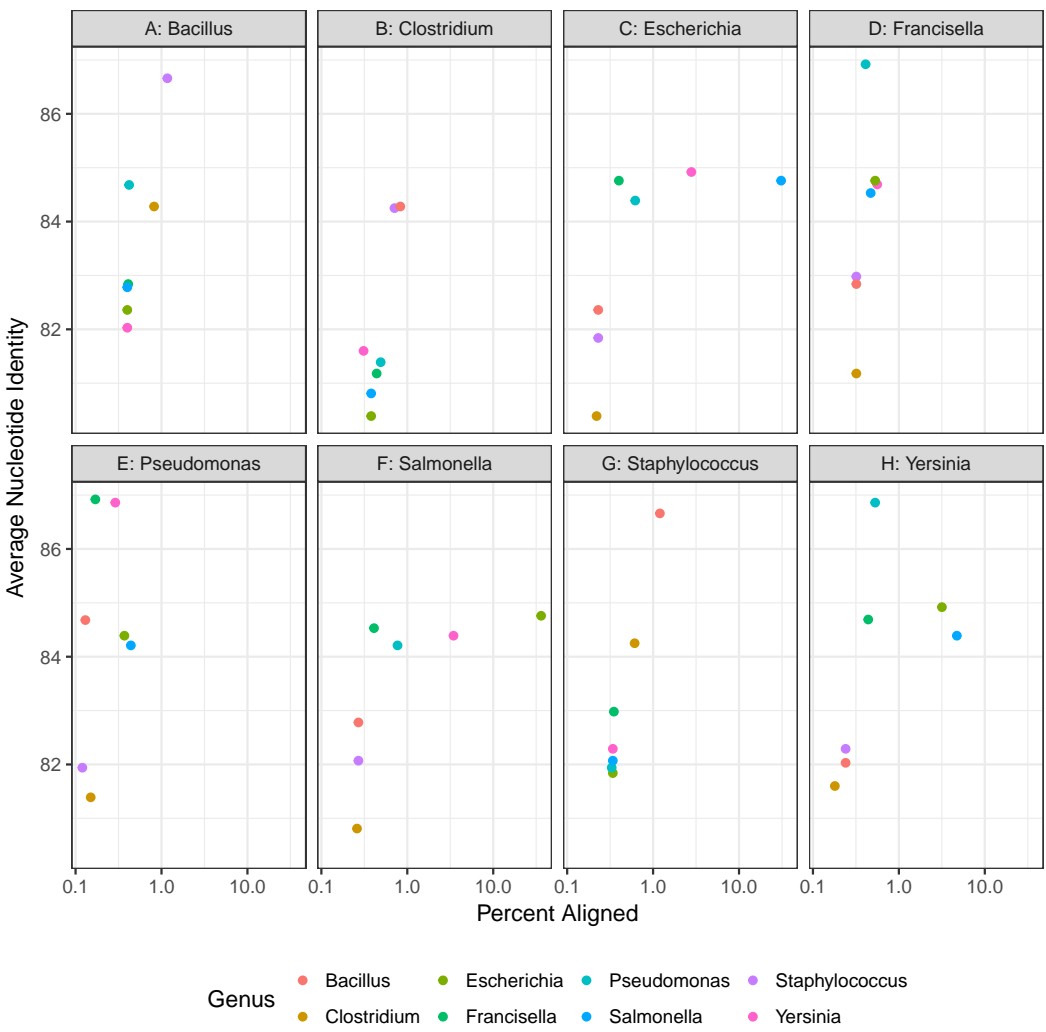

**Figure 6** Genomic similarity as percent of genome aligned and average nucleotide identity, between pairs of representative strains used in contaminant detection assessment. Plots are split by representative genomes of the material species and point colors indicate values (ANI and percent of aligned genomes) of contaminant species when aligned to the representative genome.

$Y. pestis$ was only detected at a higher proportion relative to the other datasets, $10^{-1}$ versus $10^{-3}$, and $F. tularensis$ was not detected at all. One possible reason for the lower contaminant detection for these two organisms is that there are fewer genomes in the database for these two genera. Another potential reason is that genomes from both $Y. pestis$ and $F. tularensis$ have high insertion sequence content and readily undergo intragenomic recombination (*Larsson et al., 2005*; *Chain et al., 2004*). These genomic rearrangements have been attributed to these organisms' highly pathogenic nature and challenge taxonomic classification methods due to relatively high levels of sequence similarity between seemingly unrelated organisms. Additionally, the $F. tularensis$ dataset is much smaller relative to the other genera, less than 90,000 reads. Therefore, with fewer reads in the dataset and genomes in the database, the probability that the randomly selected subset of reads spiked
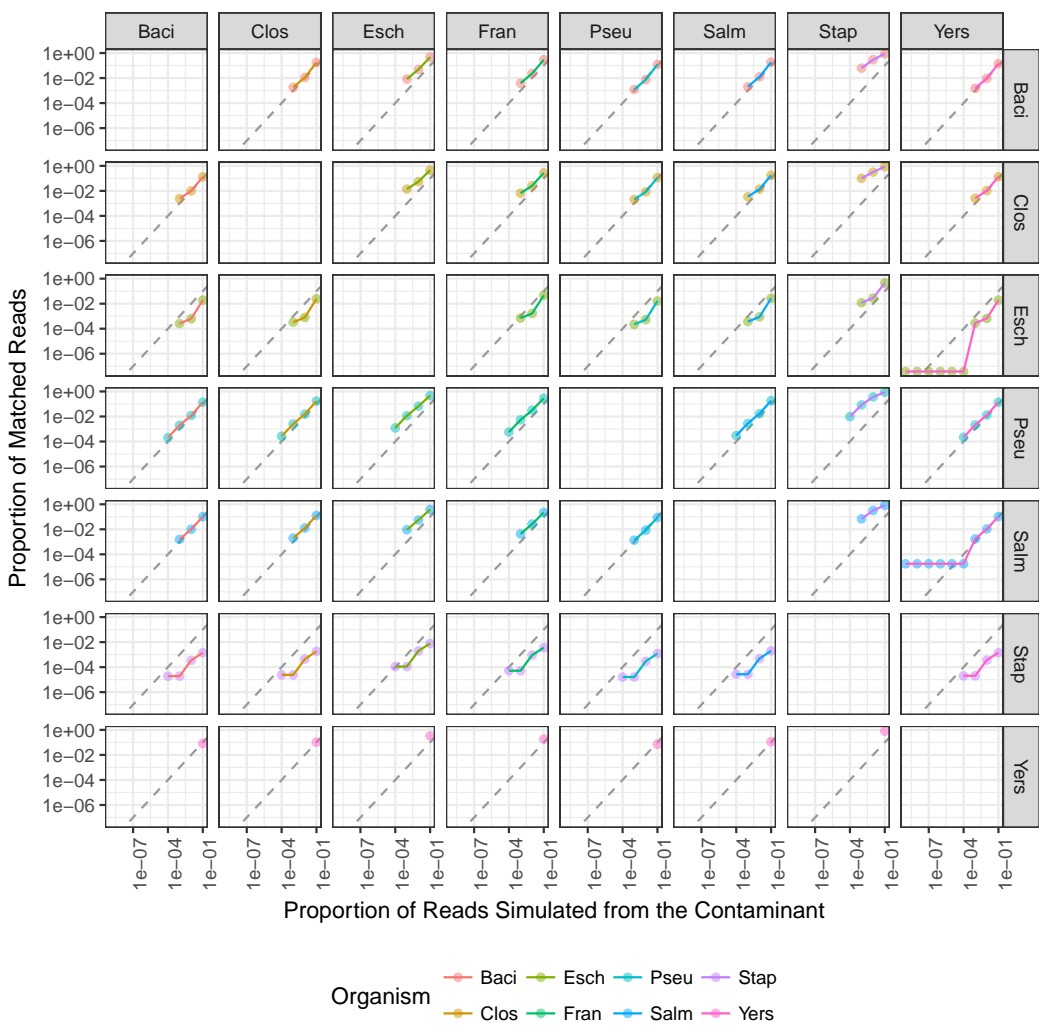

**Figure 7** **The relationship between the proportion of reads matching the contaminant species and the proportion of simulated contaminant reads.** Plots are faceted on the *x*-axis by material species and on the *y*-axis by contaminant species. Point color indicates contaminant species and line color indicates material species. Dashed line indicates the expected 1:1 correlation between the proportion of reads matching the expected contaminant and the proportion of reads simulated from the contaminant. The contaminant proportion was underestimated for points below the dashed line and overestimated for points above the dashed line.

into the simulated material dataset contains reads allowing for contaminant detection is lower. A few contaminants were detected at proportions as low as $10^{-8}$, such as when *Yersinia* contaminated with *E. coli* or *S. enterica*. However, contaminants detected at lower proportions were due to reads simulated from the material genome incorrectly assigned to the contaminant. The simulated contaminant-free *Y. pestis* material dataset had false positive reads assigned to two of the contaminants resulting in artificially low contaminant detection proportions for *Salmonella enterica* subsp. enterica serovar Typhi str. CT18 and *Escherichia coli* O104:H4 str. 2011C-3493 with estimated proportions of $1.76 \times 10^{-5}$ and $3.77 \times 10^{-8}$, respectively. The simulated dataset coverage accounts for the observed

minimum detected contaminant proportion. As the individual datasets were simulated at 20X coverage, <300,000 reads were simulated for each dataset, and on average <3 reads were spiked into the material datasets for simulated contaminant proportions $\leq 10^{-5}$ (Fig. 7).

In addition to the minimum detected contaminant proportion, we also evaluated the quantitative accuracy of the contaminant detection method. Pearson's correlation coefficient was used to determine the correlation between the estimated contaminant and true contaminant proportions for simulated contaminant proportions greater than $10^{-6}$. The estimated and true proportions were strongly correlated for all pairwise comparisons, with an overall median and 95% confidence interval of 0.99945 (0.96945–1) (Fig. 7). Eight of the pairwise comparisons have correlation coefficients below 0.99, all of which have *S. aureus* as either the contaminant or the material. Two coefficients were below 0.98: *S. aureus* contaminated with *P. aeruginosa* and *S. enterica*, 0.952 and 0.969 respectively. The total error rate was used to assess the accuracy of the PathoScope contaminant proportion estimates (Fig. 8). The material genome strongly influenced the total error rate with *E. coli* and *S. aureus* having consistently higher total error rates compared to the other genomes, indicating a reduced accuracy for the two species. The proportion of reads from *E. coli* and *S. aureus* in the simulated contaminant datasets is consistently overestimated by PathoScope (Fig. 7). Both genera had a higher proportion of false positives due to phage relative to the other genera analyzed in this study (Fig. 5). Higher phage content or mobile elements due to HGT and plasmids are potentially responsible for the overestimated proportions. In this study, the similarity between the material and contaminant genome did not impact the quantitative accuracy of the method. However, one would expect significantly lower quantitative accuracy for highly similar genomes.

## DISCUSSION

We developed an *in-silico* approach to evaluate the ability of an existing taxonomic sequence classification algorithm to detect contaminant DNA in whole genome sequence datasets from microbial materials. The use of *in-silico* data allows for a known contaminant taxonomy and concentration to challenge the algorithm. Here we used single and binary mixtures of organisms. While binary mixtures of organisms do not necessarily capture the complexity of real-world samples, they do serve as an appropriate model system to evaluate the algorithm and our approach to detecting contaminants. This approach could easily be adapted for *in-silico* datasets with multiple contaminants.

There are three basic steps to using this method to detect contaminants in a microbial material. Baseline assessment is the first step. For a baseline assessment, reads are simulated from the reference genome of the organism of interest and processed using a taxonomic classification algorithm. Performing a baseline assessment allows one to identify the false positive contaminants you can expect to observe due to methodological limitations. Simulating data with realistic error profiles, read length, and fragment distribution is likely to yield results more representative of what one would expect from real sequencing data. Next, sequencing data generated for binary mixtures of the microbial materials is processed using the same taxonomic classification algorithm as used in the baseline
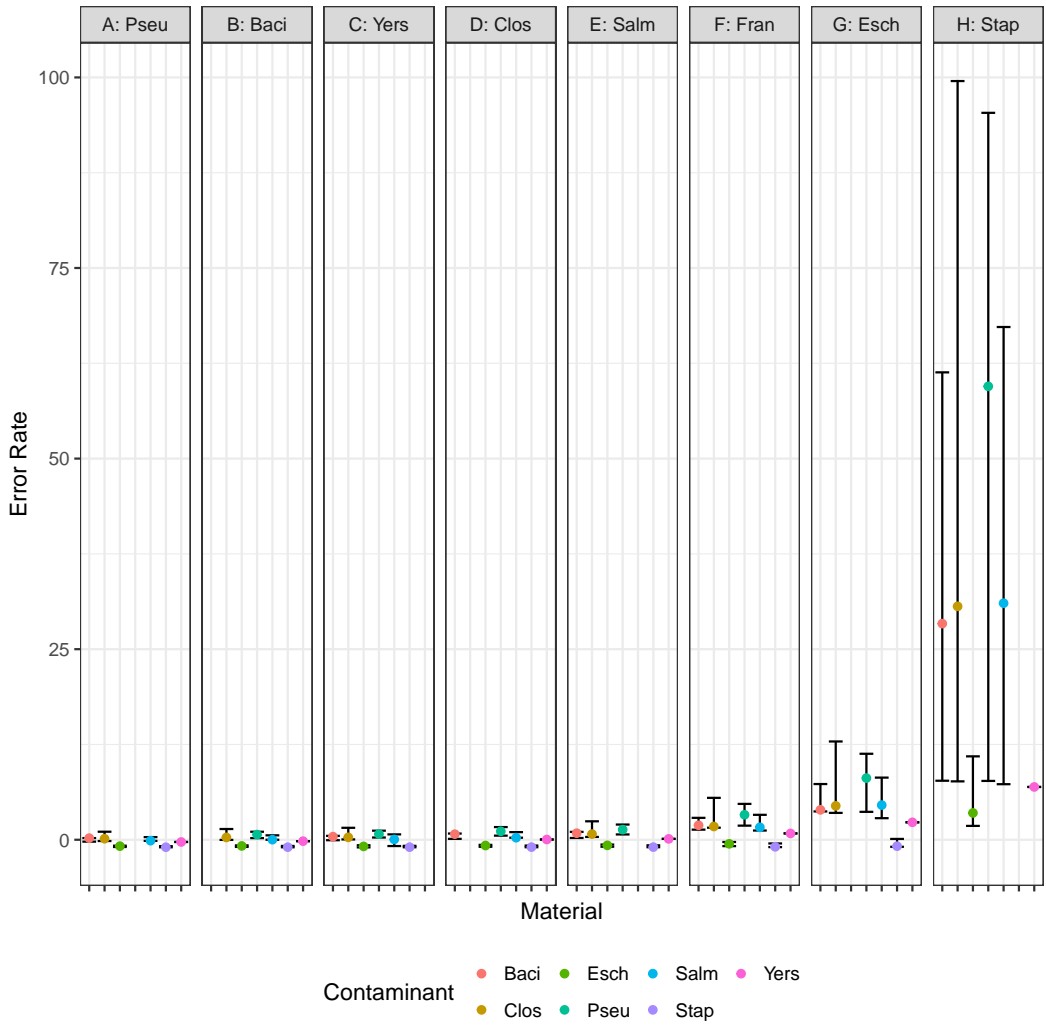

**Figure 8** Error rate, $(estimated - true)/true$, for pairwise combinations of material and contaminant. Points and error bars represent the median and range (minimum–maximum) error rate for each material and contaminant combination.

assessment. The last step is a critical evaluation of results for potential false positives. For all settings including basic research, clinical, regulatory, and attribution, the contaminant detection method should be validated for the intended application. Appropriate validation approaches may include experiments with simulated contaminants like those performed as part of this study and sequencing genomic DNA or cells spiked with varying contaminant concentrations. It is important to note that the method if routinely deployed cannot determine if the contaminants are viable and/or culturable as only the DNA is evaluated. Separate culture techniques would have to be performed in parallel to determine if the contamination was viable.

False positive contaminants identified in steps 1 and 2 were split into two categories: (1) those due to an inability of the method to differentiate the material genome from the contaminant genome, and (2) errors in the reference database. Contaminant detection
performance was characterized for different materials, contaminants, and contamination levels. Overall the method was able to identify contaminant proportions at $10^{-3}$ for most contaminant-material combinations. This level of detection is dependent on not just the classification method but also the simulated coverage. Therefore a lower detection proportion is expected for increased coverage. A contaminant proportion of $10^{-3}$ is equivalent to 1 contaminant cell per 1,000 cells in a microbial material, or 1,000 contaminant cells in 1 mL of a $10^6$ cells/mL culture. The estimated contaminant proportion accuracy for the simulated contaminated material varied by contaminant and material strain.

Quantitative accuracy in contaminant proportions is important for applications where acceptable contaminant proportion thresholds are established. For example, a microbial material with a contaminant proportion of $10^{-5}$ may be acceptable for use in an assay where the contaminant adversely impacts an assay when present in proportions greater than $10^{-4}$. Quantitative accuracy is also relevant when performing a general characterization of the microbial material. General contaminant characterization is appropriate for reference materials with more than one use case such as the microbial genomic reference materials (NIST RM8375) (*Olson et al., 2016*). Similar to the false positive contaminant baseline assessment, simulated data can be used to evaluate the minimal detectable contaminant proportion for specific organisms using different taxonomic assignment algorithms and databases. A primary limitation of the proposed method is the observed false positive contaminants identified in the baseline assessment.

The reference database and taxonomic assignment algorithm are likely to impact the number and types of false positives. There are three primary types of taxonomic read classification algorithms: sequence similarity search, sequence composition methods, and phylogenetic methods (*Bazinet & Cummings, 2012*). The example algorithm used in this study, PathoScope, is a type of sequence similarity search algorithm. Evaluating different types of algorithms using simulated data for the material genome of interest, similar to what was done in the baseline assessment part of this study, would help determine the optimal classification algorithm for a specific microbial material. Furthermore, recent advances in taxonomic classification algorithms have led to the development of faster methods, including Kaiju, a sequence composition method, and Centrifuge, a sequence similarity search method (*Menzel, Ng & Krogh, 2016*; *Kim et al., 2016*). Application of these new algorithms would lower the computational cost of the method. Similarity-based taxonomic classifications methods are not robust to horizontal gene transfer events and therefore alternative classification algorithms may be more suitable for contaminant detection than PathoScope (*Kunin et al., 2008*; *Weng et al., 2010*). Other methods such as MicrobeGPS and DUDes are alternative similarity based taxonomic classification methods developed to better handle organisms not in a reference database and are also suitable alternatives to PathoScope (*Lindner & Renard, 2015*; *Piro, Lindner & Renard, 2016*). Previous work for detecting contaminants in whole genome sequencing datasets calculate summary statistics including coverage, nucleotide composition (e.g.,%GC), and taxonomic classification of scaffolds (*Kumar et al., 2013*; *Delmont & Eren, 2016*). These methods while computationally more expensive than taxonomic classification algorithms may be better able to detect and identify microbial material contaminants. Similar studies

to the one presented here are warranted to evaluate the suitability of alternative taxonomic classification methods for contaminant detection. Incorporating baseline assessments using simulated data from single genomes into large benchmarking challenges such as the Critical Assessment of Metagenomic Interpretation could help improve our understanding of the limitations of taxonomic classification methods (http://www.cami-challenge.org/) (*Sczyrba et al., 2017*). This type of large-scale benchmarking challenge would help identify and characterize common causes of false positive classification errors. Though not investigated in this study, plasmids and horizontal gene transfer are likely a significant source of false positive classification errors. Results from such large-scale benchmarking challenges could be used to better characterize the extent to which mobile elements are responsible for false positive classification errors.

A number of the observed false positives were due to errors in the database and inability of the taxonomic classification algorithm to correctly identify the source of the sequence when it matches multiple organisms in the database. Users can generate application specific databases by preparing a custom database without sequences for irrelevant contaminants, such as phage, plasmids, vectors, multicellular eukaryotes, and genes known to undergo horizontal gene transfer in order to reduce the proportion of false positives. By excluding irrelevant contaminants and genes involved in horizontal gene transfer, sequencing reads aligning to the omitted sequences would no longer result in false positive contaminants. Methods for excluding sequence data from a reference database are dependent on the classification algorithm used. For example, user-specified sequence data could be removed from the reference database by PathoScope using the PathoDB function. Similarly, the developers of the taxonomic classification algorithm Centrifuge provide multiple databases: Prokaryotic genomes only; Prokaryotes and Viruses; Prokaryotes, Viruses, and human; as well as NCBI nucleotide non-redundant sequences. Caution should be used when removing sequences from a reference database. For example, vector sequences from contaminants in sequencing reagents, if excluded from the database, may be incorrectly classified as an organismal contaminant. Similarly, using a curated database free of misclassified and unclassified sequence data would further reduce the proportion of false positive contaminants (*Tennessen et al., 2015*). For example, the *Bacillus subtilis-Synechocystis* chimeric genome appeared to have a high false positive contaminant rate in the baseline assessment part of this study due to the genome being incorrectly classified as *Bacillus subtilis* and not a chimeric genome.

## CONCLUSIONS

Identification and characterization of low abundance contaminants in a non-targeted manner is critical for a microbial material used in high sensitivity assays such as PCR. Whole genome sequencing combined with taxonomic assignment algorithms provides a viable alternative to commonly used organismal contaminant detection methods such as culturing, microscopy, and PCR. WGS requires no *a priori* information about the contaminant and can identify common as well as unexpected contaminants.

The approach presented here is suitable for characterizing an algorithm's ability to detect organismal contaminants and could be used to compare algorithms and identify

sources of false positives for organisms of interest. Further, the algorithm could then be used to detect contaminants in actual DNA sequences from both genomic DNA and whole cell microbial materials, with the only *a priori* assumption that the contaminant is in the reference database. False positive contaminants were a primary limitation of the example system and method used herein. As false positive contaminants are database and taxonomic assignment algorithm dependent, additional work is needed to improve database curation and data authentication efforts as well as characterize taxonomic assignment algorithm performance. In summary, we have provided a straight-forward *in-silico* approach using existing datasets to challenge and evaluate the use of WGS for contaminant detection. Once a given WGS-based method and its sources of false positives are well-characterized, the method could then be applied with confidence to examine microbial reference materials and real-world samples. With the continued improvement of taxonomic classification algorithms, the expansion of reference databases, and the decline in the cost of sequencing, shotgun metagenomic sequencing provides an alternative to current methods for detecting contaminants.

## ACKNOWLEDGEMENTS

The authors would like to thank Dr. Steven Lund for his assistance in developing the study. We also thank Mihai Pop, Todd Treangen, Scott Jackson, and Jason Kralj for feedback on the manuscript. Additionally, we appreciate the comments and suggestions of the Academic Editor A. Murat Eren and the peer reviewers, which greatly improved the manuscript. Opinions expressed in this paper are the authors and do not necessarily reflect the policies and views of DHS, NIST, or affiliated venues. Certain commercial equipment, instruments, or materials are identified in this paper in order to specify the experimental procedure adequately. Such identification is not intended to imply recommendations or endorsement by NIST, nor is it intended to imply that the materials or equipment identified are necessarily the best available for the purpose. Official contribution of NIST; not subject to copyrights in USA.

### Funding

The Department of Homeland Security (DHS) Science and Technology Directorate supported this work under the Interagency Agreement HSHQPM-15-T-00019 with the NIST. The funders had no role in study design, data collection and analysis, decision to publish, or preparation of the manuscript.

### Grant Disclosures

The following grant information was disclosed by the authors:
Department of Homeland Security (DHS) Science and Technology Directorate: HSHQPM-15-T-00019.

**Competing Interests**

The authors declare there are no competing interests.

**Author Contributions**

- Nathan D. Olson conceived and designed the experiments, performed the experiments, analyzed the data, wrote the paper, prepared figures and/or tables, reviewed drafts of the paper.
- Justin M. Zook conceived and designed the experiments, analyzed the data, wrote the paper, reviewed drafts of the paper.
- Jayne B. Morrow and Nancy J. Lin conceived and designed the experiments, contributed reagents/materials/analysis tools, wrote the paper, reviewed drafts of the paper.

**Data Availability**

Github

Bioinformatics pipeline: https://github.com/nate-d-olson/genomic_purity

Data analysis and manuscript:

https://github.com/nate-d-olson/genomic_purity_analysis.

**Supplemental Information**

Supplemental information for this article can be found online at http://dx.doi.org/10.7717/peerj.3729#supplemental-information.

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
