# Peer review of "Challenging a bioinformatic tool’s ability to detect microbial contaminants using in silico whole genome sequencing data"

_PeerJ, doi:10.7717/peerj.3729_

## Round 0.1 · original submission · Major Revisions

First and foremost, I would like to thank the two reviewers of the study very much for taking the time to share their expertise, making constructive comments that will definitely improve the study, and returning their reviews in a very timely fashion.

Although both reviewers comment on the relevance of the study positively, they raise important, yet addressable concerns.

Dr. Kumar's confusion regarding the message of the manuscript will likely be experienced most readers, hence, it should be addressed thoroughly. I share Dr. Kumar's opinion regarding the need for a more specific title to set the reader expectations early on. Dr. Kumar is indeed being very kind when he suggests that missing the goal of the paper may have been on him. However, I think we can take it as an important red flag if the message is not immediately clear to a reviewer. All things considered, I believe the clarity of the manuscript would improve remarkably if you were to implement his suggestions, as well as the suggestions made by the Reviewer #2.

On the other hand, the Reviewer #2 makes important and very relevant suggestions to strengthen the study with respect to its relevance to previous work, to improve the experimental design, and to expand the description and discussion of findings and its limitations given our most current understanding.

I thank the authors for their contribution, and will be looking forward to reading their revised work.


Best wishes,
Meren.

·

Basic reporting

Dear Authors

My apologies in advance if the rest of the review sounds a bit harsh - it is very possible that I did not understand the main point of your paper because of a misconception on my part (but which may mean that other readers are likely to be confused too - and this should therefore be addressed).

Although the level of professional English is excellent, I'm afraid I didn't get the main point of the paper till well into the results.

On reading the abstract, I thought at first that the paper was proposing a new way to detect contaminants. I would suggest prefacing 'metagenomic taxonomic classification algorithm' with 'an existing'. On reading the introduction, what I understood (and again, I could be wrong, so please feel free to correct me) was that this was a way to simulate different levels of contamination in a sample, and to see if that could be detected using existing methods. If that's the case, then I would rewrite the whole introduction to make that the main point. It is possible that this paper targets readers who are not used to (current generation, rather than next-generation) Whole Genome Sequencing, and therefore the equivalence between WGS and PCR/culture/microscopy needs to be established. But I think the main point of the paper should be made first. As it stands, the introduction teases that the paper will be able to address all the issues raised, which it doesn't.

My strong suggestion would be to state the contents of the last para of the intro as early as possible to avoid leading readers down a garden path. I would also like to see a statement about who the intended audience for this paper is. Perhaps even a title change is warranted: 'A systematic evaluation of Pathoscope for detecting contaminants at different concentrations in microbial metagenomic samples'. A more specific title would immediately get rid of most of my reservations about how this paper is presented. In addition, the conclusions and limitations would make more sense as they would be clear that they applied to the specific tools and databases used in this study.

I'm aware that many reviewers are criticised for saying 'you should have written a different paper'. I hope I'm not guilty of that particular reviewerism. I just wanted to say that this suggested revised title is what I think you are trying to say, and that the paper does not make that goal clear. If that's not the case, then I apologise - and will do better at re-reading the paper.

Experimental design

The methods section was clear and well written. I have some concerns about the design that are listed below, but the language/structure was fine. I especially appreciate the docker and github repositories with all pipeline dependencies, R code, intermediate files, and outputs. I wish more papers were this thorough!

## Simulation of sequencing data.
Any reason why ART was chosen? There seem to be dozens out there eg https://www.biostars.org/p/128762/. Am not aware of an evaluation but am just curious if some are better than the rest by miles (as often happens in bioinformatics software).

## Assessment of taxononomic composition
This was a big concern - why Pathoscope? It did well in its own paper (against Metaphlan and MEGAN) but has there been an independent evaluation? It would be good to see that cited if so. For example, 10.1371/journal.pone.0117711 suggested that Pathoscope has high sensitivity but poor precision (and a low overall F score). They could be wrong, of course, but that's why I'm wondering if there have been independent evaluations. I'm not familiar with this literature, but a quick glance suggested that there might be better candidates available. There might be other reasons for using Pathoscope (the paper mentions two - but I think both of those are true for other tools as well), eg: ease of installation, regular use in clinical settings, etc, but those should be listed and justified.

Also, as the results mention - it seems hard for pathoscope to distinguish between genomes with very similar sequences as it uses sequence similarity (albeit with probabilistic alignment, bayesian reassignment, and an expectation maximization reassignment). In addition, as the results mention, some genera are harder to detect because they have fewer reference genomes in the public databases. In real world scenarios, other tools that use read-depth and sequence composition as well would likely be much better (eg Anvi'o, Concoct, etc. I don't know the lit well enough to provide a comprehensive list), or tools like DUDes that use the deepest uncommon descendent instead of the lowest common ancestor (10.1093/bioinformatics/btw150) to accurately identify contaminants at low levels, or some combination of these tools.

## Baseline assessment
This was an excellent idea. If we know that some % of a genome is always mis-assigned, then we know how to interpret those results better.

## Contaminant Detection Assessment
Sampling reads at different genomic proportions is another great idea and provides a systematic assessment of the detection capabilities of this particular software. But I'm very concerned that pairwise comparisons don't capture the complexity of real world samples which could have dozens of contaminants, including non prokaryotic contaminants.

Validity of the findings

Please see my concerns regarding the overall point and experimental methods. I would like to see these addressed before commenting more on the findings in detail.

186: I agree that the false positive results can be explained by potential errors in the database, but that makes me think the choice of a similarity-based tool for contaminant detection/metagenomic profiling is not a great idea in the real world.

Additional comments

As I said at the start, I could have misinterpreted the point of this paper. It starts of sounding like a general review/methods desc of how to identify contaminants, but it quickly becomes a description of a specific tool and how it deals with levels of contaminants. The discussion/conclusions/limitations are valuable, but don't seem to be in line with the intro/paper title.

I'm happy to be corrected of course - I could have missed a key line or two. But if the point is indeed what I suggested as an alternative title, then the introduction/discussion/conclusion sections have to be completely restructured to bring this out.

Best wishes,

Sujai Kumar
The University of Edinburgh

Reviewer 2 ·

Basic reporting

The manuscript of Olson et al. describes a study aiming at evaluating the correctness of a metagenomic taxonomic classification approach (PathoScope) to detect and identify a contaminant in a genomic dataset. The work relies exclusively on simulated bacterial whole genome datasets. Two main questions are addressed: (i) the type of false positive contaminants that might be found (ii) the influence of the concentration of contaminant on the ability of the method to detect it. I find these two questions originals and relevant but I have concerns regarding some aspects of the methodology and I regret that the origin of false positive contaminants was not more investigated.
The Introduction only present experimental biology approaches (culture, microscopy, PCR) and doesn’t point out previous bioinformatics approaches that address the question of contaminants detection in a genomic dataset, see for example Kumar et al. Frontiers in genetics 2013 or Delmont and Eren Peer J 2016.
Some claims have to be moderated, for example: line 57 “All methods require a reference database”. There are recent works based on binning co-abundant genes across a series of metagenomic samples that enables discovery of new microbial organisms without reference databases (see for exemple Nielsen et al. Nature Biotechnology 2014).

Experimental design

Regarding the method, I notice several key points to address before publication:
- Simulation for individual genomes:
o 9 genera (like indicated line 99) or 10 genera (like in table 1) included?
o Why do the Supplementary table 1 include 1677 entries and 2 times the same entry? Since seed number changes between duplicates, I guess that 2 read datasets were generated for each chromosome, but this is not indicated in the manuscript. I suggest adding in supplementary table 1: a tag for plasmids and the GC percent of each chromosome. Since plasmid and chromosomes exhibit distinct genomic features I would have systematically distinguished main chromosomes and plasmids in the study.
o I suggest adding in table 1 the complete taxonomic classification of each considered bacterial genus. An indication of the genus content and diversity (number of distinct species, number of complete sequenced genomes, an indication of genome diversity) would also be very valuable.
o I don’t understand why Escherichia and Shigella genera are considerate separately whereas it is well known (and pointed out line 114 by the authors) that the corresponding species are not phylogenetically resolved.
- Contamination detection assessment:
o Supplementary table 2: it is not clear for me why only two strains (Escherichia coli O157:H7 str. EC4115 and Yersinia pestis CO92) include respectively two and three plasmids. Do the other strains not include plasmids? To my experience plasmids may exhibit very different genome features compared to the main chromosomes, and consequently may be sources of many biases in contaminant detection. At least I would have separated plasmids from the main chromosome in the analysis for these two species.
o I would find useful to add a global comparison of genomic divergence (like ANI, see Konstantinidis et al. App. Env. Microbiol 2006 http://www.ncbi.nlm.nih.gov/pubmed/16980418) between pairs of representative and contaminant strains.

Validity of the findings

I have several remarks regarding the findings
- Simulation for individual genomes:
Authors indicate 3 sources of wrong species identification: (1) phylogenetic similarity, (2) phage shared by distinct species and (3) errors in the reference database. Regarding phylogenetic similarity, three genera are concerned: Escherichia-Shigella (why haven’t they be merged before?) and Clostridium. Phage content seems to be a major problem in the Staphylococcus genus (figure 3) but I have doubt because the median genome phage content in this genus seems to be above 0.05 (even if 3 genomes contain more than 0.15% of phage reads) and I’m not sure it is the only source of false species assignments. To my knowledge Staphylococcus strains (especially in Staphylococcus aureus species) exhibit very similar genomes that may be also a source of errors for correct species identification. Another source of errors that is not a all discussed in the presence of plasmids, repeats and horizontally transferred regions that raise problems for species identification. Finally the third source of problems for species identification is errors detected in the reference database. This is observed for genomes of Bacillus and Pseudomonas genera.
- Contamination detection assessment:
o Results reveal two species that raise problems as contaminants: Y. pestis (detected as only 10-1 proportion while other contaminants are detected at 10-3) and F. tularensis (never detected). Authors claim that the origin of these problems is that corresponding genus include less species in public databases. I don’t agree with this statement. To my opinion there are many other sources of biases, for instance the high Insertion Sequences content and intra-genomic recombination in strains of the Y. pestis species (see for instance Chain et al. PNAS 2004 DOI: 10.1073/pnas.0404012101) and in Francisella tularensis (see Larsson et al. Nature Genetics 2005 http://www.nature.com/ng/journal/v37/n2/full/ng1499.html). Another sources of biases not discussed in this study is the possibility of horizontal transfers between those species.
o Bacillus and Clostridium contaminants curves are not visible in figure 4
o The quantitative accuracy estimation indicated that two species (E. coli and S. aureus) exhibiting systematically high error rate whatever the contaminant species but doesn’t provide any clear analysis or interpretation of this result. I noticed that E. coli included two plasmids and S. aureus several phages but this is not pointed out as a possible error source in this context.

Additional comments

I find this study interesting and relevant but I regret that (i) a curated database should have be used to infer correctness of metagenomic classification algorithm, (ii) plasmids should have be processed separately (iii) few investigation according to other likely sources of biases for contaminant detection (repeats content, low divergence level between species, presence of Horizontal transfers,…). These features were neither evaluated nor discussed.
Overall I'd like to have more results and discussion in this paper to support the interpretation.
To conclude I think that this article needs complementary efforts to reinforce its value for the microbial genomics research community.

---

## Round 0.2 · accepted · Accept

Dear Authors,

Both reviewers recognize the significant improvement, and make minor suggestions.

Instead of sending it back for another round of revision, I am accepting the manuscript, and request you to consider these pints during proof-reading stage to fasten the process.

One of your reviewer's kindly mentioned that they couldn't respond as fast as they could due to traveling obligations, and I wanted to relay that information to you. I thank both reviewers for their effort, and you for revising your manuscript.


Best wishes,

·

Basic reporting

No comment

Experimental design

No comment

Validity of the findings

No comment

Additional comments

Dear Authors

Many thanks for addressing my concerns and those of Reviewer 2. You have admirably clarified the purpose of the paper.

A few very minor points still remain however (but I don't need to see the revised manuscript)

76-78 Bioinformatic methods have been developed and used to detect contaminant reads in a whole genome sequencing dataset but to our knowledge have not been used to detect contaminants in a microbial material (Kumar et al., 2013; Delmont and Eren, 2016).

It's very kind of the authors to cite our paper and the academic editor's paper about detecting contaminants, but I would argue that every whole genome sequencing microbial metagenomics project is in effect a "contamination" detection and identification project and there are many many tools and studies on this topic. Could this sentence be rephrased and a better reference used (perhaps a review)?

Now that I understand the goal of this paper more clearly, could this sentence be changed to reflect that the method used for individual genome assessment was Pathoscope:

187: We applied our method to simulated sequencing data from individual genomes

Could 99% be converted to 0.99 as the rest of this para uses proportions, not percentages:

199: ... greater than 99% ...

Best wishes,

- Sujai

Reviewer 2 ·

Basic reporting

No remark, the manuscript was substantially improved according to our comments.

Experimental design

Several corrections were performed and additional informations were provided.
Figure 3 is very interesting and illustrates intra-species and inter-species (i.e. intra-genus) variability for the 10 considered genera.
Figure 6 is also very informative and illustrates the difficulties for detection of Salmonella/Coli contaminants. To improve understanding I suggest adding in legend that plots are split by representative genomes of the material species and point colors indicate values (ANI and percent of aligned genomes) of contaminant species when aligned to the representative genome.
My only other concern was that I didn’t success in opening Supplementary Tables 1 and 2 (I guess it is not a correct csv format).

Validity of the findings

No additional comment.